# GRODIN: Improved Large-Scale Out-of-Domain detection via Back-propagation

## Abstract

Uncertainty estimation and out-of-domain (OOD) input detection are critical for improving the safety and robustness of machine learning. Unfortunately, most methods for detecting OOD examples have been evaluated on small tasks while typical methods are computationally expensive. In this paper we propose a new gradient-based method called GRODIN. The proposed method is conceptually simple, computationally cheaper than ensemble methods and can be directly applied to any existing and deployed model without re-training. We evaluate GRODIN on models trained on CIFAR-10 and ImageNet datasets, and show its strong performance on various OOD ImageNet datasets such as ImageNet-O, ImageNet-A, ImageNet-R, ImageNet-C.

## 1 Introduction

Many problems are solved with machine learning techniques today. Prominent examples of supervised learning power could be found in many domains: image classification, speech recognition, medical surveys, self-driving cars, text translation (6; 19; 28; 24; 1), etc.. In some applications, it is very important to be confident on the prediction of a trained model: self-driving cars, medicine, stock market trading, etc.. Usually, the prediction mechanism remains uncovered and it requires additional efforts to ensure the model quality when working with real-world data. We search for a new way to automatically understand that prediction on this particular sample is not reliable with data uncertainty estimation or knowledge uncertainty estimation (2).

One of the most important tasks in this field is out-of-domain detection. It allows us to separate the data suitable for inference from the examples that are not intended for model usage. Due to the importance of the task out-of-domain methods evolved from 'naive' approaches to sophisticated algorithms. These algorithms either analyse the prediction model results to understand if we could rely on them, or build a new dedicated model for OOD. The first type of the methods we call "lightweight", as they usually could be used for on-the-fly verification of prediction. The other class could deliver better detection quality, but requires substantive computational resources and separate training. We refer to this one as "heavy".

The "lightweight" models are usually built on top of certain prediction tasks, such as classification, and are based on the properties of loss functions used in the training process. Softmax probability is an example of such property, it is directly used for many practical implementations (10; 7). The prediction quality of the softmax probability thresholding could be extended with input preprocessing and probability calibration as shown in ODIN(20) algorithm. This method provides the state-of-the-art results on most available benchmarks (tiny-ImageNet (26), CIFAR-10 (16), CIFAR-100 (16), MNIST (3), SVHN (25), etc.). However, there is room for improvement. There is a bunch of more challenging problems constructed on top of ImageNet(26) image classification task: ImageNet-A (12), ImageNet-O (12), ImageNet-R (8), ImageNet-C (9).

The alternative "heavy" methods were developed: outlier exposure (11); mahalanobis distance-based confidence score (18); class of strong and theoretically justified out-of-distribution detectors called ensembles (2; 23; 22). All these methods usually provide better results, but are hard to use in practice: some of them require changing the training loss function and therefore neural network retraining (11); or require solving auxiliary logistic regression problem and heavy computation of mahalanobis distance in (18); sometimes researcher need additional training and inference of several neural networks simultaneously for ensemble methods (22).

Recent studies in deep learning suggest that the gradient of the loss function taken for a particular example carries a lot of information other than model prediction. Influence functions (14) use information about first and second derivatives with respect to network weights to estimate what is the impact of a single precedent on test set quality. Neural tangent kernel (13) gives us another point of view on the training process and generalization from the position of gradient analysis. To the best of our knowledge, there is only one work in out-of-distribution detection domain that utilize the gradient of the model with respect to weights (17). This approach uses information from the gradient of different layers as a building blocks for their separate DNN model. The approach requires a lot of out-of-distribution data to train the model and could not be assigned to "lightweight" models.

In this paper, we focus on developing a new simple and computationally lightweight approach for detecting out-of-distribution examples based on gradient analysis. We investigate the derivative of cross-entropy loss function and decompose it into two parts, showing the difference between them. The proposed method is being tested on several challenging benchmarks and compared with various methods. Also, we analyse the impact of highly popular input preprocessing and give intuition why this technique helps OOD detectors.

## 2 METHOD

| | |
|---:|---|
| $\Gamma$ | is a statistical population. |
| $X = \{(\mathbf{x}_i, y_i)\}_1^m$ | training set, consisting of $m$ i.i.d. precedents from population $\Gamma$. |
| $\mathbf{x}_i \in \mathbb{R}^n, y_i \in \mathbb{Y}$ | $i$-th precedent features vector and corresponding label. |
| $\theta \in \Theta$ | $\theta$ is particular element of model parameters space $\Theta$. |
| $\ell(\theta, \mathbf{x}, y)$ | loss function. |
| $L = \frac{1}{m} \sum_{(\mathbf{x}, y) \in X} \ell(\theta, \mathbf{x}, y)$ | emperical loss function used in training process. |
| $h_c(\mathbf{x}, \theta)$ | output of neural network on class $c$ before softmax layer. |
| $\|\cdot\|_2$ | $l_2$ norm. |

Table 1: Notation

Model mistakes come from different sources: irreducible errors, low model capacity, wrong modeling assumptions, sample distribution shift, and so on. For each type of errors, we define a formal problem setup and develop methods to deal with it. Those setups are not mutually exclusive and their methods could complement each other.

This paper focuses on a particular problem: estimation of the probability that a particular sample comes from a different population. To formally define this problem we use notation from Tab. 1. Given this notation, there exist a statistical population $\Gamma$. Elements of this population are pairs $(\mathbf{x}, y)$, where $\mathbf{x} \in \mathbb{R}^n$ is a sample, $y \in \mathbb{Y}$ is a label. We assume that there exists a deterministic correspondence between population precedent and corresponding label. Statistically representative training set $X$ is used to train prediction model by minimization of empirical loss function $L$. With enough data and mild assumptions on loss function, the resulting model with parameters $\hat{\theta}$ should replicate correspondence between $\mathbf{x}$ and $y$. As a result, our model should give a correct prediction to any $\mathbf{x}$ from the general population. Thus, given assumptions, there is a natural way to define $\mathbf{x}_i$ as the out-of-domain sample:

$$\mathbf{x}_i \text{ is out-of-domain} \leftrightarrow \forall (\mathbf{x}, y) \in \Gamma : \mathbf{x}_i \neq \mathbf{x} \tag{1}$$

Our task is to estimate, if the particular sample $\mathbf{x}_i$ is out-of-domain, given only the estimated model parameters $\hat{\theta}$.

The core idea of our method is quite simple. We train prediction model by minimisation of some additive loss function $L = \frac{1}{m} \sum_{(\mathbf{x}, y) \in X} \ell(\theta, \mathbf{x}, y)$. When minimisation is performed by stochastic gradient descent or similar methods, the expectation of the model parameters gradient norm reduces on training set elements with the number of optimization steps. The expectation aims to zero for flexible enough decision functions. In tangent kernel space (NTK for neural models) the optimization process looks like seeking for optimization parameter $\theta$ that pulls representation of training set points $\nabla \ell(\theta, \mathbf{x}, y)$ to zero vector. We can use this property and determine the model behavior for an example by the distance of its representation in the trained tangent kernel space (defined by $\hat{\theta}$) from zero.

Using this information we can predict many practical properties of the particular prediction: expected accuracy, model uncertainty, out-of-domain points detection, etc.. In this paper, we focus on the latter case and build a method for out-of-domain detection on top of this principle.

The modern definition of the domain depends on the application and we will show this difference in the experimental section. From a theoretical perspective, we equate domain and the population. This way we allow a researcher to define their notion of the domain with the selection of sampling technique. The ideally trained model brings all points of the population to a stationary point in model parameters space. In this paper we make an assumption that kernel of this transformation is *equal* to the population and the observed transformation differs from ideal one with Gaussian error: $\nabla_\theta \ell(\theta, \mathbf{x}, y)|_{\hat{\theta}} = \nabla_\theta \ell(\theta, \mathbf{x}, y)|_{\theta^*} + \delta, \delta \sim \mathcal{N}(0, E)$. Now we can define a log probability of population $\Gamma$ membership:

$$g(\mathbf{x}, \hat{\theta}) = logP(\mathbf{x}|\Gamma) \sim \left\| \nabla_\theta \ell(\theta, \mathbf{x}, y)|_{\hat{\theta}} \right\|_2^2$$

The main problem for practical application of the method is to define a loss function for examples, that come to the model during its exploitation and have no label. In this case, it is natural to use the class assigned by the model $\hat{c}$ as a label and check the uncertainty of the model in the decision. We focus, without loss of generality, on image classification problem over $k$ classes with neural models. Negative log likelihood over softmax probability modeling is one of the most commonly used loss functions for this problem:

$$\ell(\theta, \mathbf{x}, y) = -\log \frac{e^{h_y(\mathbf{x}, \theta)}}{\sum_{c=1}^k e^{h_c(\mathbf{x}, \theta)}}$$

where $h_c$ are outputs of the neural model for each class $c$. Using this model and the label assumption we can formulate the result log probability of population membership $g(\mathbf{x}, \hat{\theta})$ for arbitrary point $\mathbf{x}$:

$$g(\mathbf{x}, \hat{\theta}) = \left\| \nabla_\theta \log \frac{e^{h_{\hat{c}}(\mathbf{x}, \theta)}}{\sum_{c=1}^k e^{h_c(\mathbf{x}, \theta)}} \Big|_{\hat{\theta}} \right\|_2^2$$

Now we can choose significance level $\alpha$ and filter out points that belong to the population with probability less than this significance level: $P(\mathbf{x}|\Gamma) < \alpha$. In practice we don't use significance level directly, but define a threshold $\delta(\alpha)$ for $g(\mathbf{x}, \hat{\theta})$. If we have information on out-of-domain examples we choose the threshold that delivers the selected significance level on cross-validation. We use this setup as a baseline variant of our method in the experimental section.

In the literature there are at least two ways how to enhance the result of the baseline model: temperature scaling (5), and input perturbation (20). In our framework these enhancements are easily explained and compliment the baseline model. To show this we need to unfold components of the gradient vector:

$$
\begin{aligned}
\nabla_\theta \ell(\mathbf{x}, \hat{\theta}) &= -\nabla_\theta \log \frac{\max_{c'} e^{h_{c'}(\mathbf{x}, \theta)}}{\sum_{c=1}^k e^{h_c(\mathbf{x}, \theta)}} \Big|_{\hat{\theta}} \\
&= S(\mathbf{x}, \hat{\theta}) G(\mathbf{x}, \hat{\theta}), \text{ where} \\
S(\mathbf{x}, \hat{\theta}) &= \frac{1}{1 + \sum_{c \neq \hat{c}} e^{(h_c(\mathbf{x}, \hat{\theta}) - h_{\hat{c}}(\mathbf{x}, \hat{\theta}))}} \\
G(\mathbf{x}, \hat{\theta}) &= -\sum_{c \neq \hat{c}} e^{(h_c(\mathbf{x}, \hat{\theta}) - h_{\hat{c}}(\mathbf{x}, \hat{\theta}))} \frac{\partial (h_{\hat{c}}(\mathbf{x}, \theta) - h_c(\mathbf{x}, \theta))}{\partial \theta} \Big|_{\hat{\theta}}
\end{aligned}
$$

In the last expression there are two multiplicative parts ($S$ and $G$). $S$ is a scalar part, that reflects the probability of the predicted class. The $G$ is cumulative gradient vector for all classes output nodes of the network weighted by their preference over predicted class $e^{h_c(\mathbf{x}, \hat{\theta})} / e^{h_{\hat{c}}(\mathbf{x}, \hat{\theta})}$. The first part is equal to the *softmax score* from ODIN up to the notations used. The second part is the information from NTK space.

As one can see both $S$ and $G$ heavily depend on class weights $h_c$, and it is important to calibrate these values for the task the model is used for. One of the most popular ways of model calibration is

temperature parameter injection (5). In this method we scale class weights $h_c$ with the temperature factor $\frac{1}{T}$. The optimal temperature is then optimized using some calibration quality metric like Expected Calibration Error (5) for the target dataset. The solution of this optimization problem depends on the data provided, see the experimental section for details.

The other way of out-of-domain detection model improvement is altering the source signal, e.g. the image in case of the image classification task. In our framework we assume that during training process the optimization pulls domain space to a stationary point in model parameters space. As only training examples are observed, we pull the entire domain using the train set as mount points. In this process the rest of the domain slacks and this cause well-known difference in gradients between training set points and validation points. To reduce this slack we can move point towards training points. We follow the alteration procedure introduced in the ODIN paper:

$$\mathbf{x}_p = \mathbf{x}_o - \varepsilon \, \text{sign} \left( -\nabla_{\mathbf{x}} \log \frac{\max_{c'} e^{h_{c'}(\mathbf{x},\theta)/T}}{\sum_{c=1}^{k} e^{h_c(\mathbf{x},\theta)/T}} \bigg|_{\mathbf{x}_o} \right)$$

During this procedure, source image is altered to reduce the gradient and compensate the slack between training points and the rest of the domain.

To sum it up, the proposed method is a composition of $\varepsilon$-input perturbation, temperature calibration by scaling outputs by $1/T$ factor and decomposition of $\nabla_\theta \ell(\mathbf{x}, \hat{\theta})$ into two multiplicative parts: $S(\mathbf{x}, \hat{\theta})$ and $G(\mathbf{x}, \hat{\theta})$. These parts can be used as a single estimator: $g(\mathbf{x}, \hat{\theta}) = \|S(\mathbf{x}, \hat{\theta})G(\mathbf{x}, \hat{\theta})\|_2^2$, or alone: $g(\mathbf{x}, \hat{\theta}) = \|G(\mathbf{x}, \hat{\theta})\|_2^2$, $g(\mathbf{x}, \hat{\theta}) = S(\mathbf{x}, \hat{\theta})$. The last variant gives us exactly ODIN.

## 3 EXPERIMENTS

We performed a set of experiments to study the properties of our method, its effectiveness and its competitiveness with other approaches. Both versions of the proposed gradient method were evaluated: $G$ and $S * G$ to show that there is a significant difference in their performance.

We used popular image collections as a source of out-of-domain examples. Benchmark is standard: out-of-distribution detector is evaluated as a binary classifier that should separate in-domain samples from out-of-domain. The test dataset is a mixture of the in-domain test set and the out-of-domain dataset. The tiny-ImageNet dataset was used as an out-of-domain for all experiments with networks, trained on CIFAR-10. For neural networks, trained on ImageNet-1K, we used four recently proposed out-of-distribution collections, that represent different distribution shifts: ImageNet-O, ImageNet-R, ImageNet-A, and part of ImageNet-C. Image databases statistics are shown in Tab. 2.

- ImageNet-O (12) is a subset of ImageNet-22k that do not intersect with ImageNet-1K. It consists of images that fool Maximum Softmax Probability out-of-distribution detector (10) based on ResNet-50 neural network and represents distribution shift that brings new unobserved classes. Popular out-of-distribution detection methods work poorly on this setup as they utilize logits or softmax probabilities that are heavily biased by the design of the dataset.

- ImageNet-R (8) contains images of ImageNet objects with different textures and styles: paintings, sculptures, origami, cartoons, toys, etc. Classes are a subset of original ImageNet classes. This dataset represents distribution shift that does not change the semantic of objects but change their shape or texture.

- ImageNet-A (12) contains images of ImageNet objects that are extremely hard to be classified correctly by any neural network trained on ImageNet. It represents natural adversarial examples: it is obvious for humans what is depicted on an image, but trained model struggles to give the correct answer.

- ImageNet-C (9) contains copies of the ImageNet validation set that were corrupted with various noises. Original dataset consists of fifteen noise types, each noise type has five levels of severity. We selected the most challenging for the ResNet-50 model noise type according to the original paper: level five 'Frosted Glass Blur'. ImageNet-C dataset represent a distribution shift that occurs when for example a camera or transmission channel is broken.

Table 2: List of used datasets

| Dataset | Train size | Validation size | Test size | Outlier for | Reference |
|---|---|---|---|---|---|
| CIFAR-10 | 50000 | 500 | 9500 | — | (15) |
| ImageNet-1K | $\approx 1.28 * 10^6$ | 3000 | 47000 | — | (27) |
| tiny ImageNet | — | 500 | 9500 | CIFAR-10 | (27) |
| ImageNet-O | — | 600 | 1400 | ImageNet-1K | (12) |
| ImageNet-R | — | 600 | 29400 | ImageNet-1K | (8) |
| ImageNet-A | — | 600 | 6900 | ImageNet-1K | (12) |
| ImageNet-C | — | 3000 | 47000 | ImageNet-1K | (9) |

In our experimental study out-of-distribution detectors are applied to popular neural network architectures trained on CIFAR-10 or ImageNet-1K collections, such as ResNet-18 and ResNet-50. The detailed information about the models used for each setup could be found in Tab. 3. All experiments were performed with the PyTorch framework, due to a blind-review we attached source code to supplementary materials and will provide the link to the repository with the camera-ready paper version.

Table 3: List of used neural networks

| Architecture | Training set | Number of parameters | Accuracy (test set) | Reference |
|---|---|---|---|---|
| Wide-ResNet-28-10 | CIFAR-10 | $\approx 36.5M$ | 96.2 | (29) |
| ResNet-18 | ImageNet-1K | $\approx 11.5M$ | 69.7 | (6) |
| ResNet-50 | ImageNet-1K | $\approx 25.5M$ | 76.1 | (6) |

The following out-of-distribution detectors were used as competitors to our method:

1. Maximum Softmax Probability (MSP), also referred as a baseline, described in (10). This method uses negative maximum softmax probability as an out-of-distribution score.

2. ODIN described in (20). This method is a modification of MSP. It combines input preprocessing and MSP together using negative maximum softmax probability of preprocessed image as an out-of-distribution score.

3. Maximum Logit (ML) described in (7). This method uses negative maximum logit (neural network output without applying softmax activation function) as an out-of-distribution score.

4. Ensemble method(2; 23), trained with different random seeds. The ensemble consists of 6 different uncertainty measures that can be used as a scoring function (confidence, entropy of expected, expected entropy, mutual information, EPKL, MKL) and we choose the one that brings the best separation in terms of ROC-AUC as the final score. Here we need to train several models from scratch, thus the access to a training set is mandatory. As a result, the method can't be used on an arbitrary pretrained model so we evaluate it only for the ResNet-50 model, as we train it from scratch.

ROC-AUC is our quality measure for out-of-distribution detection. All table cells express experimental results in $\mu \pm \sigma$ pairs, where $\mu$ is the average value of the metric and $\sigma$ is its bootstrap deviation computed on 100 resamplings.

The rest of the experimental section consists of two parts: the competitive comparison in Sec. 3.1 and ablation study for our method in Sec. 3.2.

## 3.1 COMPARISON WITH OTHER METHODS

The following methodology is used to compare out-of-domain detectors for each collection from Tab. 2:

- The classification model is trained on a base dataset (CIFAR-10 or ImageNet-1K)

- The OOD model parameters are tuned on the validation subset of the collection to maximize the ROC-AUC score

- The resulting method is then applied to test part of the collection and reported in the result table

Parameters tuning is required for the proposed gradient method and for ODIN. We used grid-search to estimate optimal $\varepsilon$ and $T$. The grid for $\varepsilon$ was 21 evenly spaced points from 0 to 0.004. We select among $\{1, 2, 5, 10, 20, 50, 100, 200, 500, 1000\}$ to find optimal temperature $T$. The same grid for $\varepsilon$ and $T$ was used in ODIN paper (20).

Our experiments can be divided into two groups:

1. Ones that use pretrained weights from PyTorch collection. A researcher can skip the training procedure and just evaluate methods on downloaded data. These experiments were performed with the ResNet-18 model trained on the ImageNet classification task.
2. Experiments with models trained from scratch. In this work we train WideResNet-28-10 for CIFAR-10 classification and ResNet-50 for ImageNet classification. All parameters for the training procedure were taken from original papers for these models. For ResNet-50 experiments, we trained four (as proposed gradient methods require two forward and two backward passes) different models from different random seeds to evaluate ensemble methods.

Results of out-of-distribution detections are presented in tables 4, 5. One can see that both gradient methods: G and SG outperform all baselines on CIFAR-10 vs tiny ImageNet benchmark with a significant margin. Confidence intervals for G and SG parts intersect, so we can say that both methods show nearly the same performance.

Results for ImageNet-like datasets are more various. Let's inspect results for every dataset separately (both architectures ResNet-18 and ResNet-50 behave nearly the same):

- **ImageNet-O:** It is the only dataset on which SG-part is better than G-part. Still, both gradient methods outperform all baselines with giant margin.
- **ImageNet-R, ImageNet-C:** G-part outperforms all baselines on these two datasets. SG-part showed bad performance giving up to ODIN and ML.
- **ImageNet-A:** On this dataset G-part outperforms ODIN and ML but failed to overtake ensemble baseline.

To sum it up, G-part can significantly outperform all "lightweight" baselines (ones that do not require training several models from scratch) on both benchmarks. And only in the case of ImageNet-A with ResNet-50 as a predictor the ensemble model was able to outperform the proposed method.

Table 4: Wide-Resnet-28-10. CIFAR-10 vs tiny-ImageNet. ROC-AUC values.

| MSP | ODIN | ML | G-part | SG-part |
|---|---|---|---|---|
| $0.954 \pm 0.001$ | $0.97 \pm 0.001$ | $0.965 \pm 0.001$ | $\mathbf{0.983} \pm 0.001$ | $0.981 \pm 0.001$ |

Table 5: ImageNet-like datasets. ROC-AUC values.

| | Dataset | MSP | ODIN | ML | Ensemble | G-part | SG-part |
|---|---|---|---|---|---|---|---|
| ResNet-18 | ImageNet-O | $0.484 \pm 0.006$ | $0.628 \pm 0.007$ | $0.599 \pm 0.006$ | — | $0.76 \pm 0.006$ | $\mathbf{0.804} \pm 0.006$ |
| | ImageNet-R | $0.776 \pm 0.001$ | $0.84 \pm 0.001$ | $0.838 \pm 0.001$ | — | $\mathbf{0.852} \pm 0.001$ | $0.833 \pm 0.002$ |
| | ImageNet-A | $0.818 \pm 0.003$ | $0.848 \pm 0.002$ | $0.843 \pm 0.002$ | — | $\mathbf{0.853} \pm 0.002$ | $0.837 \pm 0.002$ |
| | ImageNet-C | $0.943 \pm 0.001$ | $0.974 \pm 0.0001$ | $0.974 \pm 0.0001$ | — | $\mathbf{0.985} \pm 0.0001$ | $0.95 \pm 0.001$ |
| ResNet-50 | ImageNet-O | $0.47 \pm 0.006$ | $0.599 \pm 0.007$ | $0.564 \pm 0.008$ | $0.612 \pm 0.006$ | $0.735 \pm 0.006$ | $\mathbf{0.769} \pm 0.006$ |
| | ImageNet-R | $0.803 \pm 0.002$ | $0.863 \pm 0.001$ | $0.863 \pm 0.001$ | $0.856 \pm 0.001$ | $\mathbf{0.877} \pm 0.001$ | $0.856 \pm 0.001$ |
| | ImageNet-A | $0.839 \pm 0.002$ | $0.872 \pm 0.002$ | $0.861 \pm 0.002$ | $\mathbf{0.885} \pm 0.002$ | $0.877 \pm 0.002$ | $0.859 \pm 0.002$ |
| | ImageNet-C | $0.937 \pm 0.001$ | $0.969 \pm 0.0001$ | $0.969 \pm 0.0001$ | $0.976 \pm 0.0001$ | $\mathbf{0.982} \pm 0.0001$ | $0.958 \pm 0.001$ |

## 3.2 ABLATION STUDY

The proposed method consists of three parts: input perturbation, temperature scaling, and scoring based on a gradient with respect to neural network weights. Our initial intuition suggested that a gradient-based scoring scheme is self-sufficient and could be used as a stand-alone discrepancy

measure. Turns out this is not the case, and both preprocessing steps are also essential to achieve competitive results.

We evaluated this impact on ImageNet-like datasets. This stand-alone discrepancy measure is a specific case of our final methods, thus it is enough to set free parameters for particular values: $T = 1$ and/or $\varepsilon = 0$. Thus we compare the final model with 3 simplified variants: $T$-scaling only model, where $\varepsilon$ was fixed to zero and temperature $T$ was tuned with grid-search; $\varepsilon$-perturbation only, with temperature $T$ fixed to 1 and $\varepsilon$ tuned on the grid; the pure gradient-scored method with $\varepsilon$ equal 0 and $T$ equal 1. The presented results are obtained on the test set, while hyperparameters were chosen depending on the best performance on gridsearch set.

The results are presented in Tab. 6, 7. As one can see for both G and SG parts vanilla and only $\varepsilon$ tuning are nearly the same and both are weaker than only T tuning or full method. Only T tuning in some cases match the full scheme, but to achieve outstanding results it is necessary to use both preprocessing steps.

Table 6: Ablation study for G-part. ROC-AUC values.

| | Dataset | G-vanilla | G-(only $\varepsilon$) | G-(only T) | G |
|---|---|---|---|---|---|
| | ImageNet-O | $0.517 \pm 0.007$ | $0.517 \pm 0.007$ | $0.678 \pm 0.008$ | $\mathbf{0.76} \pm 0.006$ |
| ResNet-18 | ImageNet-R | $0.786 \pm 0.001$ | $0.786 \pm 0.001$ | $0.839 \pm 0.001$ | $\mathbf{0.852} \pm 0.001$ |
| | ImageNet-A | $0.826 \pm 0.003$ | $0.826 \pm 0.003$ | $\mathbf{0.852} \pm 0.002$ | $\mathbf{0.853} \pm 0.002$ |
| | ImageNet-C | $0.951 \pm 0.001$ | $0.951 \pm 0.001$ | $\mathbf{0.985} \pm 0.0001$ | $\mathbf{0.985} \pm 0.0001$ |
| | ImageNet-O | $0.502 \pm 0.007$ | $0.502 \pm 0.007$ | $0.657 \pm 0.008$ | $\mathbf{0.735} \pm 0.006$ |
| ResNet-50 | ImageNet-R | $0.813 \pm 0.002$ | $0.811 \pm 0.002$ | $0.863 \pm 0.001$ | $\mathbf{0.877} \pm 0.001$ |
| | ImageNet-A | $0.849 \pm 0.002$ | $0.849 \pm 0.002$ | $\mathbf{0.877} \pm 0.002$ | $\mathbf{0.877} \pm 0.002$ |
| | ImageNet-C | $0.948 \pm 0.001$ | $0.948 \pm 0.001$ | $\mathbf{0.982} \pm 0.0001$ | $\mathbf{0.982} \pm 0.0001$ |

Table 7: Ablation study for SG-part. ROC-AUC values.

| | Dataset | SG-vanilla | SG-(only $\varepsilon$) | SG-(only T) | SG |
|---|---|---|---|---|---|
| | ImageNet-O | $0.518 \pm 0.007$ | $0.538 \pm 0.007$ | $0.691 \pm 0.006$ | $\mathbf{0.804} \pm 0.006$ |
| ResNet-18 | ImageNet-R | $0.786 \pm 0.001$ | $0.79 \pm 0.002$ | $0.787 \pm 0.002$ | $\mathbf{0.833} \pm 0.002$ |
| | ImageNet-A | $0.826 \pm 0.002$ | $0.82 \pm 0.002$ | $0.82 \pm 0.002$ | $\mathbf{0.837} \pm 0.002$ |
| | ImageNet-C | $0.931 \pm 0.001$ | $0.937 \pm 0.001$ | $0.937 \pm 0.001$ | $\mathbf{0.95} \pm 0.001$ |
| | ImageNet-O | $0.503 \pm 0.007$ | $0.503 \pm 0.007$ | $0.659 \pm 0.007$ | $\mathbf{0.769} \pm 0.006$ |
| ResNet-50 | ImageNet-R | $0.813 \pm 0.002$ | $0.811 \pm 0.002$ | $0.811 \pm 0.001$ | $\mathbf{0.856} \pm 0.001$ |
| | ImageNet-A | $0.849 \pm 0.002$ | $0.849 \pm 0.002$ | $0.846 \pm 0.002$ | $\mathbf{0.859} \pm 0.002$ |
| | ImageNet-C | $0.948 \pm 0.001$ | $0.948 \pm 0.001$ | $0.939 \pm 0.001$ | $\mathbf{0.958} \pm 0.001$ |

## 4 DISCUSSION

The task for out-of-domain detection is usually related to sensitive information and it is extremely hard to find real-world data on this topic. To resolve this issue there are joined academic and industry initiatives such as NeurIPS Challenge (21). The data from this event is not yet publicly available and we have to use state of the art datasets, that are artificially designed. In this section, we want to analyse this bias and its influence on the methods from the experimental section.

### 4.1 MODEL CALIBRATION

In theoretical Sec. 2 we noted, that calibration procedure is mandatory because the datasets used for out-of-domain detection could differ from the training set in many ways. We use Expected Calibration Error (5) as a measure of temperature fit for our data:

$$ECE(T, X, \hat{\theta}) = \sum_{b=1}^{B} \frac{|X_b|}{|X|} \left| acc(X_b|\hat{\theta}, T) - conf(X_b|\hat{\theta}, T) \right|$$

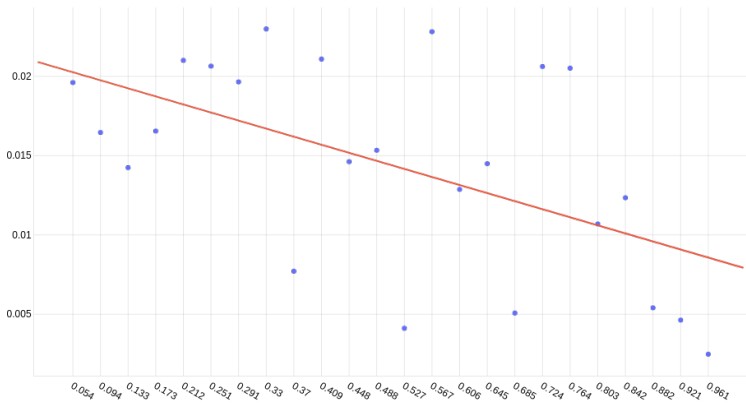

Figure 1: Dependence graph between $conf(X_b|\hat{\theta}, T)$ and $acc(X_b|\hat{\theta}, T)$ for ImageNet-A dataset. Red is the trend line.

where dataset $X$ is ordered by model confidence $\ell(\mathbf{x}, \hat{\theta}, T)$ and then split into bins $X_b$. For each bin we use two statistics: $acc(X_b|\hat{\theta}, T)$ — mean accuracy for this bin of the model with weights $\hat{\theta}$, $conf(X_b|\hat{\theta}, T)$ — mean model confidence score $e^{\ell(\mathbf{x}, \hat{\theta}, T)}$.

When the labels from out-of-domain dataset match those from the training set, this parameter can be calculated and optimized directly. During this optimization we have found, that the higher temperature $T$ is used, the better $ECE$ score becomes. This result matches conclusion from the ODIN paper, but looks counter-intuitive at the first glance. In the Fig. 1 we presented a dependence graph between $conf(X_b|\hat{\theta}, T)$ and $acc(X_b|\hat{\theta}, T)$ for ImageNet-A dataset. From this graph, it is clear, that with the growth of model confidence its accuracy becomes lower. Calibration assumes these variables are correlated which is not the case. The problem comes from dataset construction where we choose the examples, that are hard to detect (12). This way we bias the dataset towards the examples that give high model confidence but a wrong label. In this case calibration procedure become irrelevant and will choose the highest possible temperature $T$.

From our experiments, we conclude that it is practical to use higher temperatures when we use $S$ part only. Unlike ODIN, GRODIN uses information from $G$ part which is influenced by the temperature as well. With the grows of the temperature the contribution of gradient vectors of $h_c$ from all classes become even. Irrelevant classes give almost random gradients and this signal becomes too noisy. That is why we need to seek for optimal temperature. On the other hand, the signal from $G$ is based not only on class weights but on fundamental properties of the image, that are detected on different layers of the model (4). That is why our model works better on the biased towards hard examples datasets, such as ImageNet-O.

### 4.2 GRADIENT COMPONENTS

Our method is based on assumption that classification model parameters gradient contains information on point resemblance to the domain the model is trained for. But it is unclear how this information is distributed among $S$ and $G$ parts of the gradient. In this section we want to confirm the assumption that $G$ part is necessary for the model.

To understand if the $G$ part compliments information from $S$ (that is already used in ODIN) we take the in-domain and out-of-domain examples mixture from ImageNet-R that is hard to split by $S$ and use information from $G$ part for this task. In the left part of the Fig. 2 reader can find the histograms of two distributions grouped by levels of $S$: red for out-of-domain examples and blue for those in-domain. From these examples we chose those bins that are placed between two black lines. This area is the hardest for separation and with $S$ signal alone we can achieve $0.708 \pm 0.002$ ROC-AUC.

The right part of the Fig. 2 contains the histogram for this same subset of "hard" examples but from $G$ perspective. As one can see, these points are still hard to separate, but the $G$ part contains a bit different signal and allows to get $0.735 \pm 0.002$ ROC-AUC. This fact demonstrates, that information

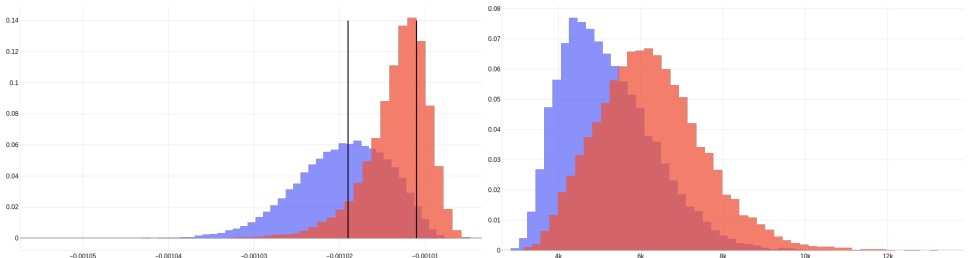

Figure 2: Histograms of ImageNet-R scoring functions. S for full dataset to the left and G for only "hard" examples to the right

from $G$ part contains additional information to $S$ and explains the difference of GRODIN results the from original ODIN algorithm.

### 4.3 METHOD COMPLEXITY

Our method is computationally quite lightweight. Input perturbation requires one forward pass through the network, and one backward pass to compute a gradient. After this, we need one more forward and backward pass to compute gradient $l_2$-norm at the perturbed point. The total complexity is $O(|F| + |B| + |\Theta|)$, where $O(|F|)$ is forward pass complexity, $O(|B|)$ is backward pass complexity, and $|\Theta|$ is a number of parameters in neural network. In practice, the computation time is approximately two times slower than single training step (forward + backward pass), and approximately $4 - 6$ times slower, than a single inference step.

## 5 CONCLUSIONS

In this paper, we have found a way to utilize a back propagation procedure for out-of-distribution detection. We think this solution is both practical and theoretically found. From a practical point of view modern DL inference accelerators are suitable for back propagation step by their design and our approach could be easily implemented in practice. The other practical advantage of the proposed method is that it uses only prediction model properties and there is no need for training an additional model for out-of-distribution detection. The theoretical basis of the proposed method comes from recent theoretical results in neural tangent kernels and allows us to say that the method will be relevant for bigger models that emerge in recent papers.

We have found that interaction of the model and the argument item depends severely on the gradient with respect to model parameters at this item. This naive and straightforward assumption allows us to predict out-of-distribution items. In this paper, we have shown that a method based on this property is able to outperform the state-of-the-art methods. We used several modern collections (CIFAR-10/100, ImageNet-O, ImageNet-R, ImageNet-A, part of ImageNet-C, tiny-ImageNet) to explore the properties of our method and to compare it with state-of-the-art results. We found that in all cases our approach gives comparable results and in eight of nine experiment cases performs significantly better.

All modern out-of-distribution detectors use parameters tuning for better detection on different collections and we found that parameters could dramatically change the performance of the method. In a situation when the environment is unknown during the training process (e.g. adversarial attacks) this property makes such methods impractical. We found that proposed method has an extremely stable set of optimal parameters that could be used in a wide variety of situations.

In this work we used a very simple approach to gradient analysis and one of the possible directions of the future work is to apply more sophisticated instruments such as gradient normalization, newton step instead of gradient, gradient component distribution analysis on per layer/block/individual basis. We believe that back propagation has a wider field of applications than out-of-domain detection and going to expand this field in the future.

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
