# OpenReview forum: "GRODIN: Improved Large-Scale Out-of-Domain detection via Back-propagation"
_ICLR.cc/2022/Conference — ICLR 2022 Submitted_

### Official Review · Reviewer_gLsy · 2021-10-26

**Correctness:** 2
**Technical Novelty And Significance:** 2
**Empirical Novelty And Significance:** 2
**Recommendation:** 3
**Confidence:** 4

**Main Review:**

A positive aspect of this paper is that it evaluates OOD detection performance on the difficult task of separating ImageNet-1k from several other subsets of ImageNet.

However, overall the evaluation is severely lacking. The authors compare to only four fairly weak baselines (and one of those only partially). As far as I understand, the authors try to justify this by creating a vaguely defined dichotomy between "light-weight" and "heavy" OOD detection methods. These methods are dismissed on the basis of being "hard to use in practice" for different reasons. I would argue that the proposed GRODIN method is far less practical than, say, outlier exposure [11] or Mahalanobis distance [18] (without preprocessing steps) because they require two full forward and two full backward passes through the network, making inference far more costly. Pre-computing a Mahalanobis distance or training an outlier aware network from scratch would be far cheaper in many cases when comparing to slowing down inference by a factor of 5x. There is no good argument to not even compare to these methods. I also disagree with the authors claim in Section 4.3 that their "method is computationally quite lightweight".

Furthermore, the authors did not properly their CIFAR-10 performance even against the baselines that they did include. They only compared the ROC-AUCs against tiny-ImageNet. They should also be testing against CIFAR-100, SVHN, noise distributions etc... In addition to this, using tiny-ImageNet as a test out-distribution may be flawed anyway because there is a clear overlap in classes (frogs, dogs etc. are part of both CIFAR-10 and tiny-ImageNet). It is quite possible that their OOD methods are not picking up semantic differences but rather detect downscaling artifacts on tiny-ImageNet. This needs to at the very least be discussed in the paper.

Another issue is that the authors' method crucially relies on hyperparameter selection on the test out-distribution. This means that their methods will not perform well simultaneously on different out-distributions. Comparing the vanilla results in Table 6 and Table 7 to the MSP results in Table 5, we see that without these steps, their method barely even outperforms the trivial baseline. While similar criticisms can be made for several older papers on OOD detection (including ODIN [20]), this issue has been circumvented by many more recent works, that do not select separate hyperparameters for each test out-distribution (e.g. [30,31,32]).

On top of these issues in the evaluation, the paper is unfortunately also quite hard to read. The paper would benefit greatly from having a native English speaker carefully edit the writing. Because of this, I found it somewhat hard to follow the authors' motivation for their method.

Minor comments:
-Table 1 caption should be above the table
- all equations are missing punctuation
- for formatting "log" or "acc" in math mode, one should use \mathrm{}
- Table 5 is quite hard to read because for some reason the error bars appear to use a larger font than the actual values
- I do not understand why the authors only used the ensemble baseline in a single setting. Their argument on page 5 that they can't use the pre-trained model seems very weak. Why not simply train the models yourself?
- It seems odd that they use a much larger model on CIFAR-10 than on ImageNet
- Figure 1 should at least have axis labels

[30] Towards neural networks that provably know when they don't know, Alexander Meinke, Matthias Hein, at ICLR20
[31] Outlier exposure with confidence control for out-of-distribution detection, Papadopoulos, Aristotelis-Angelos and Rajati, Mohammad Reza and Shaikh, Nazim and Wang, Jiamian, in Neurocomputing21
[32] ATOM: Robustifying Out-of-Distribution Detection Using Outlier Mining, Chen, Jiefeng and Li, Yixuan and Wu, Xi and Liang, Yingyu and Jha, Somesh, at Joint European Conference on Machine Learning and Knowledge Discovery in Databases 2021

**Summary Of The Paper:**

The paper proposes a method for detecting out-of-distribution examples by applying the ideas of the popular ODIN method to different scoring functions. They ablate the effects of different parameters and compare their methods to several baseline methods in OOD detection.

**Summary Of The Review:**

I recommend rejecting this paper, mainly because there are severe flaws in its evaluation.

---

### Official Review · Reviewer_zSjc · 2021-11-03

**Correctness:** 2
**Technical Novelty And Significance:** 2
**Empirical Novelty And Significance:** 2
**Recommendation:** 5
**Confidence:** 4

**Details Of Ethics Concerns:**

No ethics concerns.

**Main Review:**

The paper discussed an important problem and proposed a simple yet interesting solution. Please see detailed comments below.

Minor issues:
- better to add numbers to equations
- better to use "[]" to cite references, and keep "()" for equation indexes.
- page 2, last paragraph, "with the number of optimization steps": not clear. Maybe you want to say "when the optimization converges"?


Major issues:
 - Page 2, "This expectation aims to zero for flexible enough decision functions": this is a very strong assumption. Could you define what are "flexible enough decision functions"?
- Page 2, last paragraph, "zero vector": do you mean the norm of the vector is close to zero? or every dimension of the vector is zero? Did you assume any batch normalization or layer normalization?
- Page 3, 2nd paragraph, "the observed transformation differs from ideal one with Gaussian error": I am not sure whether "the Gaussian distribution" is the case. Could you give some empirical analysis? Or add some references? Especially I don't know if it holds in high dimensional space.
- Page 3, the 2nd equation: what if the new sample comes from a "non-of-above" category?
- Page 4, the 1st equation: Please add references to the ODIN paper. Also it will help the audience to understand if you could summarize the differences between your method and ODIN.
- Experiments are interesting, but I am curious whether the proposed method can work for binary classification, like "human vs non-human", where the out of domain example can be a dog image which category never appears in the training sets.

**Summary Of The Paper:**

This paper proposed a new method called GRODIIN to detect whether a new sample is similar with training set or from out of distribution. The basic assumption is that the expectation of gradient descent should be close to zero vector on the in-domain training set. Then this paper can compute the gradient on a new example, and compare its distance in the tangent kernel space.

**Summary Of The Review:**

This is an interesting topic and I hope to see more discussions of the proposed method. Currently I am not 100% convinced by the assumption and I wonder whether there are even better solutions.

---

### Official Review · Reviewer_CsE6 · 2021-11-05

**Correctness:** 3
**Technical Novelty And Significance:** 1
**Empirical Novelty And Significance:** 1
**Recommendation:** 3
**Confidence:** 5

**Main Review:**

The idea of using gradients for OOD detection is interesting but not new. This work does not cite any related works in this direction. Several papers show up quickly with a simple search; none of these previous works are discussed in this paper. ([3] is relatively new, so it is ok to not include it in this submission)

[1] https://ieeexplore.ieee.org/abstract/document/9190679
[2] https://arxiv.org/pdf/2007.09507.pdf
[3] https://arxiv.org/abs/2110.00218

This work has several issues in comparison with previous works and the justifications of the proposed method:
1. There is no explanation for why the norm of gradients can be used to estimate the logP(x|gamma) and how good the estimation is. The first equation on page 3 has no proper justification.
2. How is the proposed method compared to previous methods that also use gradients (see [1][2][3])? Besides, the text misses a proper citation in many places. For example, at the top of page 3, there should be citations when saying the gradients can predict expected accuracy, model uncertainty, and out-of-domain detection.
3. The paper regards ODIN as the SOTA and uses it as the primary baseline for the comparison. But it is not true, and there are several light methods potentially better than it. For example, Mahalanobis [4], Gram matrix [5], GODIN [6] are also light by the definition of the paper. A reasonable experiment section should at least compare to one of the above methods.
4. Would you please discuss or analyze why high-temperature T helps to generate the gradients that comprise a better OOD score?
5. Section 2 reads not organized. Please consider dividing it into sub-sections and adding indices to the equations.
6. The customed citation format is confusing. Please consider using the ICLR official format.

[4] https://arxiv.org/abs/1912.12510
[5] https://arxiv.org/abs/1807.03888
[6] https://arxiv.org/abs/2002.11297


**Summary Of The Paper:**

This paper proposes to use the norm of gradients as the score to detect OOD data. The experiments use CIFAR-10 and ImageNet as the distribution datasets and use the unseen classes or perturbed images for the OOD data. The result shows the norm of gradients improves the OOD detection.

**Summary Of The Review:**

This submission lacks a proper justification for the designed method and lacks an appropriate comparison to related works. These issues signal a rejection.

---

### Official Review · Reviewer_cmXa · 2021-11-05

**Correctness:** 3
**Technical Novelty And Significance:** 1
**Empirical Novelty And Significance:** 1
**Recommendation:** 3
**Confidence:** 4

**Main Review:**

The novelty and technical rigor of this paper are rather low.

Regarding novelty, there have been several score-function-based OOD detection procedures already proposed, and this work does not refer to any of them.  For instance, [Grathwohl et al. [ICLR 2020]](https://arxiv.org/abs/1912.03263) propose $- \mid\mid \nabla_{\mathbf{x}} \log p(\mathbf{x}; \boldsymbol{\theta}) \mid\mid_{2}^{2}$, with the primary difference being the derivative formulation.  More related still is the work of [Sharma et al. [UAI 2021]](https://arxiv.org/abs/2102.12567).  They use the Fisher information for OOD detection: $$E_{y} \left[ (\nabla_{\theta} \log p_{\theta}(\mathbf{y} | \mathbf{x})) (\nabla_{\theta} \log p_{\theta}(\mathbf{y} | \mathbf{x}))^{T} \right].$$

This paper needs to justify why the proposed variant is superior to these alternatives, both conceptually and empirically.

**Summary Of The Paper:**

This paper proposes an anomaly detection score based on the derivative of the log-likelihood: $ \mid\mid \nabla_{\theta} \log p_{\theta}(\hat{c} | \mathbf{x}) \mid \mid_{2}^{2} $ where $\hat{c}$ is the predicted class (as we want the method to be applicable to test points).  The intuition is that if $\mathbf{x}$ is from training set, then the model should already be fairly well fit to the point, meaning that the derivative should be near zero.  This metric along with some variants derived from sub-terms are compared on image data sets against baselines such as ODIN, maximum softmax probability, maximum logit, and ensembles.

**Summary Of The Review:**

The paper does not properly discuss, compare to, and improve upon related work that also uses OOD scores of the form $\frac{\partial}{\partial \ \cdot} \log p_{\theta}(\mathbf{y} | \mathbf{x})$.

---

### Decision · Program_Chairs · 2022-01-20

**Decision:**

Reject

**Comment:**

The paper proposes a gradient-based method for OOD detection. While the paper has some interesting contributions, all the reviewers felt that the current version falls below the ICLR acceptance threshold. I encourage the authors to revise and resubmit to a different venue.